# Examining Conjoint Behavioral Consultation to Support 2e-Autism Spectrum Disorder and Gifted Students in Preschool with Academic and Behavior Concerns

**DOI:** 10.3390/bs13080674

**Published:** 2023-08-11

**Authors:** Gül Kahveci, Ahmet Güneyli

**Affiliations:** Faculty of Education, European University of Lefke, Mersin 10, Lefke 99728, Northern Cyprus, Turkey; gkahveci@eul.edu.tr

**Keywords:** conjoint behavioral consultation, family-school partnerships, academic skills, listening skills and on-task behavior

## Abstract

Conjoint behavioral consultation (CBC), as adjusted for 2e children with academic and behavioral difficulties, was the focus of this single-subject design study. Three young children from a preschool participated, together with their parents and teachers. Academic enablers for students—intervention, maintenance, and generalization phases; academic and behavioral competencies—intervention, maintenance, and generalization phases; and teachers’, parents’, and students’ perceptions of the intervention’s social validity data were collected among the outcome measures. Findings from multiple participant-related probes pointed to constructive improvements in the phases of intervention, maintenance for listening behavior, and improved on-task skill in intervention, maintenance, and generalization. Additionally, during the consultation, parents and teachers noted improvements in the outcomes of the target behavior such as rhythm keeping, picture–word matching, writing the pictured concept in Turkish, writing the pictured concept in English, short personal story writing, short personal story telling, and verbal math problem solving, and each stakeholder gave the intervention a grade for its social validity. Limitations, potential routes for future study, and implications for preschool CBC intervention are highlighted.

## 1. Introduction

Finding the interventions and behaviors that can best be used to enhance quality of life for neuro-divergent people or people with autism is becoming increasingly important to researchers, teachers, and families [1]. Some of the current applications include applied behavior analysis interventions, the use of assistive technology, and video modeling [2,3,4]. A person with autism may experience social, physical, or emotional limitations as part of a spectrum known as autism spectrum disorder (ASD). The neurodevelopmental disorder ASD is frequently characterized by communication and social interaction difficulties. One such classification of ASDs is high-functioning autism; like ‘mild’ autism, high-functioning autism is an unofficial term that has gained wider acceptance. Teachers frequently mention students who are enrolled in two programs—one for the gifted, and one for special education as ‘twice-exceptional’ (2e) learners. These individuals have a very high degree of intelligence and have a thorough comprehension of the relationships between and computation of words, numbers, music, physics, or any other specific topic of study. Compared to other categories of autistic people, these outstanding individuals are less common. However, it should be highlighted that extra care must be made to establish a mentally stimulating atmosphere that is sufficient to allow these unusually endowed youngsters to develop and to learn how best to apply their skills.

One definition of ‘giftedness’ is the recognition of the ‘extraordinary person’. Gifted children categorized by this definition are quite uncommon. Roughly one in 100,000 or one in a million, depending on the part of the world, are labeled as such according to anecdotal statistics. A gifted child may occasionally encounter substantial obstacles when attempting to express their ability in conventional academic settings [5].

Another important idea is ‘twice exceptional’. Despite students with two exceptionalities, or ‘2e’, having both strengths and weaknesses, the two exceptionalities are typically treated independently in learning environments [6]. This is carried out without taking into consideration the potential impact that one exceptionality may have on another [7]. These students usually experience a specific learning disability in topics such as math, reading, or writing. Some persons may also have ASD or ADHD due to their complex personalities.

Conjoint behavioral consultation (CBC) is a system in which school staff, family members, and educational experts who provide consultation service carry out observation and problem-solving stages together. After the meetings with the school staff and family members, the learning environment, learning content, and learning characteristics of the neuro-divergent individuals are observed. The most notable difficulty encountered in the education of neuro-divergent individuals is the need for individual adaptations of teaching content and teaching strategies. One of the reasons for specially tailoring the content for children with 2e-ASD is that the existing curriculum content is weak, since the child has in-depth knowledge of the subjects learned, with the other reason being these children only having specific and limited areas of interest. Another challenge is the transfer of teaching materials to the child. It has been revealed in imaging studies that the brain mechanism that provides neural connectivity, called the short and long ‘connectom’ at the macroscopic level, which provides the transfer of information, develops differently from the students with typical development [8,9]. It should be carried out in a way that is compatible with the child’s brain network system. At this point, among the prominent elements are sharing the knowledge as concretely as possible, teaching the content by emphasizing the topics that the child shows a deep interest in, and the use of music-based teaching practices. Conjoint consultation processes are of paramount importance as providing this mechanism in teaching requires expertise.

There is scientific agreement that most autistic children, including 2e students will experience worse educational, health, and life outcomes than their neurotypical counterparts [10]. Executive functioning issues, such as issues with shifting attention, working memory, self-control, and impulsivity, as well as issues with mental flexibility and task starting (on-task behavior), can cause academic challenges in children with autism and 2e [11]. According to Mazurek et al. [12], such children may exhibit high rates of difficult behavior, which may interfere with their relationships with teachers and other pupils. Additionally, according to Levy et al. [13], up to 63% of children with autism also have a co-occurring language impairment, which may affect the child’s capacity to connect with classmates, obey school rules, and most importantly, access the curriculum [1]. The way in which these children approach tasks that require high levels of executive functioning [14], such as maintaining focus, switching between activities, and independently completing tasks that involve a multitude of steps, may be influenced by the cognitive and behavioral differences that they exhibit [15].

Generalizations made according to the information processing system of typically developing children may not be valid for children with ASD. For example, the comments in the literature reveal that children with ASD’s limited areas of interest reduce their social communication behaviors and can turn their reflections into teaching subjects out of interest by changing the educational content [16,17,18]. Contrary to this educational approach, however, there is emerging evidence to suggest that incorporating the interests of children with ASD into interventions can have positive effects [19]. The first positive results of ‘interest-based intervention’, one of the educational strategies, were obtained from the study of Koegel, Dyer, and Bell [20], who found that including children aged 4 to 13 with ASD in their preferred activities led to noticeable reductions in social avoidance behavior. In line with the findings of this study, a study by Martin and Farnum [21] on 3 to 16-year-old children with autism spectrum disorder found that including new and unusual animals in children’s intervention sessions increased social behavior and resulted in less stereotypical behavior. The positive results of this interest-based instruction in children with ASD have thus been widely reported in the literature [22,23].

One of the primary goals of education ought to be the reduction in students’ dependence on adults, alongside the promotion of independent participation and performance of classroom tasks [24,25]. This applies to students with 2e. This is crucial in determining how these children approach tasks requiring high executive functioning and independence, with the school environment a crucial context for using effective instructions such as evidence-based practices (EBPs), where children with autism and 2e spend the bulk of their time learning [26,27]. As a result, such interventions and behaviors can be used to best enhance quality of life for neuro-divergent people.

### 1.1. Structured Instruction

Structured instruction is a treatment for autism spectrum disorder (ASD) that was developed in the early 1970s as part of the Treatment and Education of Autistic and Related Communication Handicapped Children (TEACCH) program. According to Humphrey and Parkinson [28], structured instruction is recognized as a viable method substantiated by evidence and is frequently utilized in ASD-specific environments dedicated to learning. This is a method that makes use of both physical and visual structure in order to reduce anxiety and produce an environment that is ‘autism-friendly’ for individuals of varying ages and degrees of functional ability [29]. Utilizing different types of visual information is the fundamental component of this tactic.

### 1.2. Visual Timetables/Schedules

Visual schedules provide students with information regarding an anticipated sequence of events by utilizing visuals and symbols. This helps students become oriented while also creating a sense of predictability within the classroom. According to Sterling-Turner and Jordan [30], giving persons with autism spectrum disorders regular transitions helps these individuals feel more comfortable and display less troublesome behavior [31]. The fact that the National Autism Center [32] considers such schedules to be ‘established interventions’ highlights the significance of these tools in fostering a sense of autonomy in children and assisting them in the process of developing plans for their respective futures. According to the findings of a study that was carried out by Knight, Sartini, and Spriggs [31], the use of visual schedules proved to be beneficial in terms of encouraging on-task behavior and facilitating independent transitions. This categorization of schedules as ‘visual supports’ is comparable to Wong, Odom, Hume, and others’ [33] use of the same term. According to the findings of these studies, one method that is useful for aiding pupils who have autism or 2e is the utilization of visual timetables [29].

### 1.3. Conjoint Behavioral Consultation in Preschools

It is essential to provide forms of assistance such as conjoint behavioral consultation, or CBC, in understanding and addressing the needs of students with 2e for preschools. As consultation has evolved in the realm of human services, numerous manifestations of these aspects can be found in various consultative approaches. It is unsurprising that a worldwide agreed-upon definition of consultation is lacking given the multiple variations observed [34]. Studies on the effectiveness of curricula and other associated interventions to improve living conditions and overall quality of life are particular areas of focus for this research. The Community Building Conversations program is a family–school cooperation project that places equal emphasis on developing relationships between parents and teachers and basing communication in a strategy for problem-solving [34]. A consultant is responsible for directing the CBC process with families and school staff in order to realize the goals of strengthening the link between the home and the school, facilitating healthy connections, expanding the capabilities of parents and teachers in terms of parenting and teaching methods, and encouraging positive child responses. The collaborative behavior assessment is a process that involves a parent and a teacher working together over the course of three meetings to (a) identify strengths and areas of concern, (b) choose a priority or target concern that will be the subject of consultation, (c) look at factors that may obstruct or promote adaptive change, (d) create a support plan to be used at home and at school, and (e) assess progress and adjust the plan as necessary. This process can be broken down into the following steps: (a) Determining your best attributes. Collaborative behavior assessment is a personalized method that makes the most of the benefits that are already there and is based on the ecological systems theory [35]. (b) Collaborative behavior assessment and specialized care for students who have extensive support needs in schools share a number of characteristics, including a focus on adult–child interactions and constructive methods for supporting students both at home and at school. These are just two of the many similarities between the two types of programs [36].

Historically, schools have typically been able to fulfill the requirements of these students by first making an effort to address the weaknesses of the students [7] through a variety of courses, activities, and learning environments [37]. When a student’s disability or problem is given most of the attention, this subfield of psychology, known as ‘positive psychology contentions’, whereby just a portion of the student’s potential is being considered, is enacted. [38]. Instead of concentrating primarily on making improvements to one’s deficiencies, positive psychologists advocate the use of evidence-based treatments that boost one’s level of competence by highlighting and increasing one’s strengths [39]. According to the research of Seligman and Csikszentmihalyi [40], treatment involves sustaining the good aspects as well as repairing whatever is wrong. Courtesy of this method, the consultant will be able to base the guiding principles of the school and its curriculum on a theoretical framework that combines positive psychology and the talent development strategy, as has been supported by research on 2e-autism spectrum disorder and gifted education. This will allow the consultant to better serve the needs of the school’s students.

The ‘course of study’ for a specific class or cohort of students is what we mean when discussing a curriculum [41]. The curriculum is not simply a collection of enjoyable activities that take place in the preschool classroom; rather, it is a set of plans and activities that lead to the children acquiring new knowledge. These plans and activities are what comprise the curriculum. A comprehensive guide for the instruction of young children and their day-to-day interactions with adults should be included in a curriculum of this nature, and this guide should encompass all aspects of education. A modification to the curriculum is any change that is made to an activity or piece of content in the classroom in order to make it possible for a child to participate in that activity or content [42]. Material adaptations are when teachers make adjustments to the materials so that the child can engage with as little assistance as possible. The use of children’s preferences is, nevertheless, another form of adaptation. This involves adults determining what activities children enjoy and incorporating their interests into the activity in order to make it more engaging for the children [41]. There are a few aspects that need to be acknowledged when adapting the curriculum for children who have 2e. These elements include concurrent supports for the child’s academic performance, as well as their social and emotional well-being, both of which are essential to the development of the child. These simultaneous supports include things such as accommodations, therapeutic interventions, and specialized instruction. This also encompasses educational experiences that are enhanced, or advanced, and have the goal of developing the child’s interests, strengths, and talents while simultaneously satisfying the learning needs of the child [43]. Following this procedure, it will be possible to confirm that the revised curriculum has a structure that is based on strengths, is talent focused, and supports the development of talent.

The current single-subject research design related evaluation is based on the intervention including visual schedules and curriculum modifications that directly support executive function in a classroom setting, while also assisting in the general reduction of anxiety. This evaluation is related to whether the intervention package included visual schedules and curriculum modifications.

Consequently, the goal of this research was to determine the extent to which factors, such as putting CBC protocols and visual schedules into action, as directed by a workbook for teachers, would have an effect on the participants’ listening abilities and on-task (their ability to remain focused on the job) behavior. After the conclusion of the intervention, will the participants be able to keep up their listening abilities and maintain their attention on the job they are performing in the second, fourth, and sixth weeks? Will the participants be able to generalize their behavior while they are performing their task?

## 2. Methods and Materials

The method that was utilized in this case study was a multiple-probe design across all the participants. It is advised that probe procedures be used in order to establish the degree to which behavior has become independent of treatment contingencies and responsive to natural consequences [44]. The evaluation processes that are currently utilized to develop evidence-based practices for autism [45] can accommodate a multiple-probe design without any significant modifications [46]. When it comes to providing proof of an intervention’s efficacy, large group studies that employ randomized controlled trials are often regarded as the gold standard [47]. On the other hand, single-subject research methods are more suited to giving information on certain individual behavioral changes [48].

### 2.1. Setting

The investigation was carried out at a preschool where the number of pupils in each class was capped at between eight and ten. The research was carried out in three separate classes, with a single participant from each of those classes. The professional staff at the school included a school psychologist, and teachers with expertise in their own subjects (for example, experienced musicians who are Orff specialists and preschool teachers), but these teachers did not necessarily have credentials in special or gifted education.

### 2.2. Materials

Because visual schedule strategies can be used to lessen anxiety and create an atmosphere that is ‘autism-friendly’ for 2e people, a workbook resource was made available to teachers that was closely based on Haas’s writing [49]. Along with more detailed information regarding visual timetables and work systems and access to internet resources, the workbook featured material on structured teaching. It then provided instructions on designing a work system and a daily itinerary for pupils, with examples of several work systems including a checklist and organized literacy, science, math, art, and music activities (participant selected), with numbered task pieces and intervention strategies so that the curriculum can be strength based, talent focused, and allow for to talent development. Additionally, a template for a straightforward checklist that organizes and breaks down tasks was given to teachers. An implementation checklist based on the fundamental components of visual schedules, as explained in the workbook, was also included. The instructors were given the directive to follow the instructions that were provided on the checklist in order to guarantee that visual schedules were available for use by all of the participants.

Listening skills: Reading preferred books: *Hayvanlar Nerede Yaşarlar*? (*Where Do Animals Live*?) (ISBN 978-605-312-396-5).

Music: Listening to songs with ‘Habitats Song’ (https://www.youtube.com/watch?v=byvf7jwdvOI) and participating in some parts of it by keeping rhythm with various Orff instruments. (accessed on 4 July 2022).

English: Learning related words and concepts in the song ‘Habitats Song’. Matching pictures and word flashcards.

Writing/Literacy: Writing the names/concepts related to shown picture flashcards in the ‘Habitats Song’ in Turkish and English (pencil–paper activity).

Short Personal Narrative Writing: Creating a schema-based personal narrative within a structured process.

Expressive Language: Telling a short personal narrative with a simplified version [50].

Mathematics: Problem structure includes change, group, compare type verbal math problems. There are three different types of verbal mathematical problems related to the animals they prefer in their worksheets. There are pictures of the related animal on the worksheet and an empty table for the student to draw the mathematical diagram and write the numbers on it [51,52].

In Appendix A, all of the materials used in this study are explained in detail.

### 2.3. Participants

The three students who participated in preschool were included in this study—two boys and a girl aged five. The common characteristic among the gifted and autistic children who participated in the study is that they learned to read and write on their own at the age of three. Each participant’s anonymity is maintained throughout this article by utilizing pseudonyms to identify them. All of the students were Caucasian, each coming from a middle- to high-socioeconomic background. In order for a student to participate in the research, they needed to provide a comprehensive psycho-educational evaluation that included scores that attested to both high skill levels as well as documented behavior concerns. Each participant went to the school as a prospective student, took part in instructional sessions during which teachers observed their reactions and interactions, and then engaged in in-depth interviews after attending preschool for a total of two weeks. The results of standardized tests, personal interviews, and in-person observations were deliberated by the admissions committee when making their decision. Every student’s score on the Intelligence Scale for Children—WISC IV indicated that they all possessed high-intellectual capacity, as was independently validated [53]. The range of results for the verbal comprehension test was from 110 to 120, the range for the full scale test was from 120 to 139, and the range for the perceptual reasoning test was from 106 to 132. The following diagnoses were found in psychological assessments completed on participants before they took part in the study: Asperger’s syndrome (n = 3), attention-deficit hyperactivity disorder (n = 1), and generalized anxiety disorder (GAD, n = 2). Deficits in executive functioning, processing speed, working memory, and/or writing output were identified for all of the participants in the study, and these deficiencies were recorded in the reports. Before they were allowed to participate in the study, two of the persons involved had been participating in treatment and assistance programs that were tailored to fit their specific needs. One was currently taking medication, or had previously used medication, to treat a combination of inattention, hyperactivity, and anxiety. No pupil had hitherto been able to succeed in preschool environments, despite these supports being in place. Participants in the cohort were all identified as 2e, but each person’s mix of strengths and limitations was unique. The brief summaries of the participants, which were created from the folders for admissions, psychological reports, interim progress reports, and cumulative folders, give a sense of the breadth of their specific challenges and needs, as well as a feeling of the experiences that are shared among the members of the group.

Alex (Participant A): Asperger’s syndrome, generalized anxiety disorder, and (ADHD) were identified as Alex’s diagnoses. In his records, it was noted that he had extreme cognitive rigidity as well as social immaturity. On the WISC IV, he scored a 120 overall, and his perceptual reasoning score was 132, indicating that he had exceptional intellectual ability. Students at his school predicted that he would continue to struggle in social situations and in settings with sizable groups of people. Alex did not want to go to school since he claimed that his assignments were too simple. He added that it was futile to repeatedly go over the subjects he already knew, and that he would rather construct a sandcastle in the garden of the school instead, explicitly expressing this preference. He was upset that his favorite topic, wall heaters, as well as animals and dinosaurs, were not discussed at all in the course.

Olivia (Participant O): Diagnosed with Asperger’s syndrome and generalized anxiety disorder. Olivia had gone to two different therapists, and was also receiving medication for her condition. She was worried, unhappy, afraid of change, and frequently oppositional when she first started attending this preschool. Her highly varied WISC IV profile began with a score of 115 and ranged from there. The total score for the perceptual reasoning section was 106. Her resistance led to confrontations with both the pupils and the teachers in the classroom. Inadequacies in executive functioning were discovered as well. She has a hard time getting into the swing of things at school. According to the reports, the individual has challenges with executive functioning, which may contribute to anxiety, impair her capacity to govern her own behavior, and result in poor educational performance (factors that lead to anxiousness). During her time in the preschool, which lasted for a total of two weeks, she experienced significant growth in her artistic skills.

Bailey (Participant B): According to the findings of the psychological evaluations, Bailey was given a diagnosis of Asperger’s syndrome when he was 3 years old. On the WISC IV test of verbal comprehension, he had a score of 115. The total score for the perceptual reasoning section was 120. Inadequacies in motor planning and executive functioning were discovered as well. Upon starting preschool, Bailey was already seeing a special education teacher once a week to receive assistance with the difficulties he was having with writing and organization. He would not complete any written assignments, whether in school or at home.

Following the meeting where the participants’ families and teachers were consulted, the following characteristics were identified as being shared by all three. Everyone in the group is able to read and write. They are able to perform the four fundamental mathematical operations with two-digit numbers. Behavioral observations consisted of the following: daydreaming; attention deficit disorder; difficulty maintaining concentration; skills in listening; a diminished ability to listen attentively; preoccupation with one’s own ideas and concepts; apparent boredom; task completion (difficulties in completing tasks); duties that are strongly linked to personal interests; tasks that are frequently left unfinished when they are not related to strengths/motivations, avoiding prolonged mental exertion unless they are engaged in the task, and failing to persevere with jobs that appear to be pointless. The following are some other shared characteristics: disorganization; the risk of losing items that are important for the job; creating a chaotic environment; having trouble following directions; reactions to authority (questioning regulations and directives); having difficulty following instructions; impulsive behavior that is accompanied by a lack of interpersonal judgment (not waiting their turns, interrupting), and lack of ability to exercise enough self-control.

After the consultation meeting with the families and teachers, the basic stressors and main problems were determined as follows: the basic stressors stem from curricula not being aligned to a child’s strengths, styles, or interests, the inability to learn academics, inability to make friends, and inability to attend to tasks. Frustration and anxiousness are the emotional results. Behaviors such as withdrawal, avoidance behaviors, angry outbursts, inattention, hyperactivity, and impulsivity are behavioral manifestations.

The points that the consultation team should focus on are as follows: the curriculum is not adapted to the child’s preferences, strengths, or styles, the inability to complete school work, duties and make friends.

### 2.4. Procedure

#### 2.4.1. Before Proposing an Implementation Plan, CBC–Family

Curriculum adaptation takes place after a consultant–family interview about the child’s gifts, abilities, interests, preferences, strengths, or styles. The consultant explained the aim of this interview and its important derives from the understanding that such a focus provides opportunities for capturing students’ attention [54] and identifying paths for their positive development [40,55].

The consultant elaborated on the significance of adopting a strength-based perspective and its connection to individual variations in terms of learning: According to Baum and Owen [56], Gardner [54], and Silverman [57], the way in which second language learners organize their lives and process information is influenced by style distinctions. Because adopting a strengths-based perspective necessitates paying close attention, the team makes use of the students’ profiles in order to appropriately align educational goals [58].

The consultant emphasized the significance of social and emotional development [59,60]. Well-being in social and emotional domains forms the foundation for efficient learning. Students who have been diagnosed with multiple disorders usually struggle with psychosocial and emotional issues [61,62]. Both Baum, Dann, Novak, and Pruess [63] and Eide and Eide [64] state that twice-exceptional students frequently display symptoms of anxiousness in addition to social and emotional immaturity. It is possible that due to these issues, they will not be able to live up to the standards that are expected of their age or grade. In light of all of these considerations, the team gives readiness extensive thought.

The consultant stressed the significance of the family context, stating the following: families of children with autism spectrum disorder typically have a history of having contentious interactions with the communities of their children’s schools; because of a parent’s nervousness, lack of confidence in communicating with instructors and staff, and general worry about their child’s future success, a student’s academic performance and their ability to adjust to their new environment at school may suffer [56]. A further potential source of contention is when a parent’s expectations and those of the school are at odds with one another. Ongoing contact and various tools are provided to parents in order to facilitate improved support for their children outside of school and to encourage parental participation in the group that makes decisions regarding their children’s education.

#### 2.4.2. Before Proposing an Implementation Plan, CBC–School

Students had access to a range of informative services from CBC, including opportunities for talent development, acceleration, and enrichment both within and outside the classroom. The curriculum was matched to the learning profiles, interests, strengths, and preferences of the students. When coming up with solutions to an issue, the CBC team started by considering various hypotheses to shed light on the behavior. For instance, the following inquiries could be made if a student was reluctant to finish their work: Is the task challenging enough for the student? Does the student regularly have the chance to develop their talents? Is the work being performed in class carried out in two separate ways to meet both strengths and weaknesses? Can we sufficiently support the student to decrease his/her anxiety? Can we support the student enough to increase their well-being?

#### 2.4.3. Curriculum Modifications and Enrichment

Students had access to a wide variety of educational services provided by CBC, which included possibilities for talent development, acceleration, and enrichment both within and outside of the classroom setting. The students’ learning profiles, interests, and strengths were taken into consideration when designing the curriculum, and the students’ preferences were also taken into consideration.

All of this information that was communicated between the consultant and the family or school offered a consistent foundation for the ongoing development of the curriculum, education, and enrichment opportunities for each student by taking these aspects into consideration. The CBC team began the process of developing answers to a problem by first pondering a number of hypotheses intended to explain the observed behavior. For instance, if a student showed signs of reluctance to finish their assignment, the following lines of inquiry could be pursued: Is the assignment difficult enough for the student to handle? Is the learner provided with opportunities on a consistent basis to develop their skills? Is the work that is being performed in class executed in two different ways, one to accommodate strengths and the other to accommodate weaknesses? Are we able to support the learner to the extent that it will help reduce their anxiety? Are we able to provide the student with an adequate amount of help to improve their well-being?

Following the second consultation meeting with participants’ families and educators, the following qualities of the group as a whole were identified as being particularly prevalent.

Attention; having difficulty maintaining focus; daydreaming; listening skills; impaired ability to listen attentively; preoccupied with own thoughts and notions; appears bored, having difficulty sustaining concentration; daydreaming. task completion; difficulties in completing solo tasks; duties that are strongly linked to personal interests, and tasks that are frequently left unfinished when they are not related to strengths and motivation. Avoids extended mental strain unless they are engaged in it, and they do not persevere with jobs that appear to be pointless. Disorganization; the risk of losing items that are important for the job, and a chaotic environment. Having trouble following directions; inconsistently questioning regulations and directives and having difficulty following instructions. Impulsivity: impulsive behavior that is accompanied by a lack of interpersonal judgment (not waiting turn, interrupting); reactions to authority and a lack of ability to exercise enough self-control.

After the consultation meeting with family and teachers, the basic stressors and main problems were determined as follows: curriculum not aligned to child’s strengths styles, or interests; inability to learn academically; inability to make friends and inability to attend to tasks. Emotional outcome: frustration and anxiety. Behavioral manifestations: complaints about school; physical ailments; avoidance behaviors; aggressive responses; inattention and hyperactivity and withdrawal.

The points that the consultation team should focus on are as follows: the curriculum is not adapted to the child’s preferences, strengths, or styles and inability to complete schoolwork, duties and make friends.

The first author documented the procedures to be followed throughout the full study process in a control chart to support internal validity. In this procedure, certain flexible difficulties remained. For instance, changes were made to meeting dates and/or time.

Baseline for the intervention. Prior to the implementation of the intervention, the first author observed all participants a minimum of three times in order to establish a baseline for the desired behaviors. It was emphasized to the educators that they should continue with their typical class preparations. Both the behaviors of the students and the prompts given by the teacher to the target children were recorded from the very beginning of the activity, collecting data, and assigning codes to this collected data.

The initial author carried out all the data collection themselves. The ten-second partial interval coding method was used to record behaviors whenever they occurred during an observation interval, which were very intermittent. Because this recording only covered a portion of the interval, it was able to keep track of both on-task behavior within the given amount of time. A digital audio player started playing an audio recording that was 10 min long and consisted of alternating segments of music and silence, each lasting 10 s. The first author wore headphones while conducting research and took notes during periods of silence, later transcribing those notes while listening to music. The duration of each job varied. The instructor provided the signal to begin working, and continuous observations were carried out during the brief tasks. When activities were spread out over longer periods of time, observations were planned to capture student behavior at various points of the assigned tasks at the beginning, in the middle, and at the end of the activity. Unexpected delays and the classroom activities during which the students were supposed to stop working were not taken into account in the observations. Following the coding procedure, the percentage of times the participant displayed the objective behavior, using a percentage of the total number of times individuals observed, was calculated.

The phase of intervention. At the beginning of each phase of the intervention, the first author presented the instructor with a physical copy of the workbook and sat down with them in person for around half an hour to discuss the material contained within it.

### 2.5. Dependent Variables

On-task behavior patterns. At any point during an observation interval, the student was considered to be writing, typing, drawing, telling, or reading if they were holding their pencil or another writing implement, with the tip making contact with the paper, worksheet, or exercise book, telling their personal narratives, or pressing a key on a computer keyboard. Additionally, the student was considered to be reading if they were holding their book open and looking at it. On-task behavior contains many skills. When all 10 task behaviors are considered, listening skill is considered as a common denominator. For this reason, on-task behaviors were evaluated within the wider framework of listening skill. The student is required to show active participation by presenting many course content materials. First and foremost, children should listen to the requested questions and then answer them. For this reason, it was included in the single-subject research model graph as ‘correct response percentage in listening skills’ for simplicity according to the given answers, regardless of whether they were correct or not. During the course of an observation interval, these actions may occur at any point in time. For the student who was observed working independently on a wider variety of assignments, being on task included the following activities: writing, telling, cutting, coloring, or gluing, where these were the activities in which the student was expected to be involved, if the student was using scissors to cut the intended material, if his/her pencil, other coloring tools, or glue stick was held with the tip touching the paper, or if he was pressing paper with glue to the intended swatch. The student who was observed working independently on schedule was given a score after the completion of each step in the following task analysis: (a) returning the completed task activity card to a ‘finished work’ basket located next to the picture schedule (this step was not used on the first trial); (b) locating and removing the current task activity card from the schedule; (c) locating and walking to the corresponding work area; and (d) beginning the task with the appropriate materials. After the completion, a score for being on schedule was assigned. The ‘on-schedule designation’ was not awarded until each stage had been satisfactorily completed before moving on to the next.

When a student failed to successfully complete a step of the ‘on-schedule’ task analysis or took longer than 10 seconds to finish a step during the ‘on-schedule’ task analysis, an off schedule was recorded for that student.

When a student was on schedule and either (a) visually attending to the appropriate scheduled materials, (b) looking at their picture activity schedule, (c) manipulating the appropriate scheduled materials, (i.e., as they were designed to be used), or (d) transitioning from one schedule activity to another, a recording of ‘on-task with scheduled materials’ was made. It was required of the teachers that they maintain a record of the behaviors that each student displayed when they were not working on the task that had been assigned to them. The participant engaged in activities that were unrelated to the task that was to be completed, such as getting up from their chair, moving around the room, leaving the room, working on things that were not related to the task the instructor had assigned (such as drawing or using the computers), and having conversations with fellow students. Distracting behaviors include talking and refusing to work, two instances of which are given here. The operational definition of ‘off-task’ includes several problem behaviors that were applicable to all students in general. These behaviors, in addition to those that were specific to particular students, were included in the definition. These actions included yelling, throwing things, running away from the workspace or room, falling to the ground, and becoming violent toward other students or teachers (i.e., grabbing, hitting, pushing, or shouting). Any instruction, help, reminder, or cue given by the teacher, teacher’s assistant, or adult volunteer to draw the student’s attention to the task, work system, or schedule was considered to be a form of prompting. Prompting was carried out in order to ensure that the students were successful in completing their assignments. There are many different types of prompts, including (1) verbal instructions, (2) using the student’s name, (3) gesturing or pointing, (4) touching the student, (5) touching or tapping his chair or desk, (6) showing him a visual cue, and (7) using proximity to draw the student’s attention. Regarding the process of prompting, the teachers did not receive any guidance or direction from the administration. When the prompts were being recorded, a distinction had to be made between those that were provided more generally to the entire class and those that were addressed more clearly to the student (for instance, those that used the student’s name or were in response to the student engaging with the teacher, e.g., whole-class instruction). The only prompts considered were those that addressed the learner directly or responded to them in a manner that was direct. These are the written words that were writing samples obtained from students who had been assigned writing assignments, and this was accomplished by photographing or photocopying the students’ finished products. The students were given writing tasks in order to facilitate the collection of writing samples. After that, a computation was conducted to determine the total amount of writing that took place throughout the activities that took place in both the baseline and intervention phases of the research. It was impossible to obtain an accurate tally of the number of words or pages that each child had written throughout each session, simply because there was no way to track their progress. It was not always evident how many words a student added while participating in a class, and in other instances, students removed their own work prior to it being recorded, so it was difficult to ascertain how many words each student contributed. Certain writing assignments required completion over the course of multiple sessions in order to be finished.

### 2.6. Validity in Relation to Social Settings

At the conclusion of the intervention period, an email containing the survey was directed to the email addresses of the teachers. It was mandatory for them to complete the survey. The survey inquired about the usefulness of the workbook–activities–interventions format, as well as the effectiveness of the methods, the ease with which they could be applied, the level of motivation and independence possessed by the students who utilized the strategies, and whether or not the instructors would recommend the strategies to other people. The first author also conducted interviews with the students to find out how they felt about using visual timetables and work systems, how simple they found these tools to use, and how beneficial they thought these techniques had been. Specifically, the first author wanted to know how the students felt about using visual timetables and work systems. In particular, the author who first coined the idea sought to find out how the students felt about the use of graphic timetables and work systems. To carry out this interview with the student, the student was excused from class at a time that was agreeable to both the student and the interviewer. It was formatted in the form of a questionnaire, and the responders were given a list of options from which to choose.

### 2.7. Reliability

A second observer, who had been trained in the procedures of data collection and had taken part in a pilot project that had occurred before this inquiry, was utilized to evaluate the accuracy of the coding. The dependability of the coding was accordingly evaluated with the assistance of this additional observer. The second observer was given a set of written instructions that described the dependent variables in detail, as well as a sheet for making an exact replica of the observations that the first author had used. This was carried out in order to ensure consistency between the two observers’ experiences. Both observers were stationed in close proximity with one another, either at the back or side of the classroom, as they carried out their observations. This was carried out to ensure that both observers were unable to see the recording being produced by the other. It was necessary to make use of an additional set of headphones in order to play the audio track for both of the observers at the same time. Eighty percent of the sessions that were being watched were analyzed in order to produce an estimate of reliability. In every single one of the measurements, the proportion of intervals in which both observers arrived at the same conclusion was more than 87%. This was the case in every single one of the experiments.

### 2.8. Fidelity

Since CBC includes a comprehensive intervention process, fidelity includes both CBC interview processes involving school professionals and family, and teaching processes applied to participants [65,66]. It was determined that the interviews with the school professionals and families summarized the problems. The specific problems to focus on were decided upon, with the individual characteristics of each participant child being determined, and the teaching strategy and content adaptation agreed to be 100%.

The initial author used the same checklist that was given in the workbook to carry out a fidelity check. This was carried out in order to ensure that the information was accurate. The following are examples of the content of the checklist: ‘The student who is the focus of this activity has their own personal schedule, and the activity or lesson is segmented into a number of tasks and task steps to make the following points abundantly clear: (1) what is expected of the students, (2) how much is expected of them, (3) how to determine when the pupils have completed their work, and (4) what comes next’. The first author went over each session and validated everything that could be checked with visual evidence by checking it off on a list. After that, the percentage of items that could be checked off the list was computed. Except for the postponement of the two planned intervention sessions to the next day, the training process was carried out as planned, in full.

## 3. Findings

Findings of this study are presented in Figure 1, Figure 2 and Figure 3.

As a first step, the teachers shared the activity content and sequence of activities with the participants, after receiving information from the first author about the purpose and applications of visual schedules using these visual supports. Listening skill was used as the first activity. In listening skills (Figure 1), while the first participant did not react at all to the listening-related course content presented to him before the treatment, in the course content and lecture presentation, which were adapted considering the individual choices of the participants, the number of correct responses immediately after the treatment was 40, 60, 50, 60, 90, 90, and 100 percent, rising incrementally. While the second participant’s percentage of correct answers was 10 and 20 percent, it increased steadily over the course of seven sessions, rising from 60 percent to 100 percent. While the percentage of correct response in the third participant varied between 0 and 20, it reached from 70 percent to 100 percent in seven sessions after the treatment. In the continuation phase, the participants continued to respond correctly at 80–100 percent level in six to seven sessions.

Other activities include rhythm keeping, picture–word matching, writing the pictured concept in Turkish, writing the pictured concept in English, short personal story writing, short personal story telling, and verbal math problem solving (Figure 2). The correct response level percentage of the participants before and after the treatment is presented as a bar graph. Among the studies that included visual schedules and related concepts including areas of interest, the percentage of rhythm attitude reached from 40 to 90 percent, respectively. The picture is from 30 to 100 percent in word-matching skill, the concept shown in the picture is from 40 to 100 percent in Turkish writing skill, the concept in the picture is from 30 to 100 percent in English writing skill, and it is in short personal story writing skill, short personal story telling, and verbal math problem solving skills. From the level where there is no feedback, the level where the correct feedback is 100% has been reached.

The on-task skill (Figure 3), which is included in all these studies, was not observed in all participants, but a hundred percent level was reached after the treatment, and the level of 90–100 percent was maintained in both the follow-up and generalization phases.

## 4. Discussion

Visual analysis of the data was performed first, with the goals of identifying changes in level, variability, trend, overlaps, gaps between intercepts, and consistency between phases in Figure 1 and Figure 3 [67,68]. There is no tried-and-tested approach to determining how well youngsters are performing. When examining well-being, some research only employed instruments that measured negative indicators such as sadness and anxiety, whilst other studies only utilized measurements of positive indicators such as self-esteem and life satisfaction. A portion of researchers made use of a combination of these methods [69]. In this research, when the developments related to participating in classroom studies and performing tasks by showing success in eight academic fields were examined, it was observed that the participants were able to adapt to the curriculum within the framework of their own interests. It has been observed that especially when the order of the activities is given a structure known to the participant beforehand with a visual schedule, and when the participant clearly understands what the teacher wants from them, anxiety and behavioral problems are nullified by being an active participant. Accordingly, one can assume that participants regularly had the chance to develop their talents, so long as participants’ anxiety was quelled enough to increase their well-being.

The idea of being twice exceptional can provide difficulties for students, as well as for their families and the institutions that educate them. Conjoint behavioral consultation is one of the methods supported to overcome these difficulties that have been shown to be effective in enhancing student academic, social, and emotional learning outcomes. In this study, the consultant (first author) works with families and school employees to facilitate the CBC process, with the goals of improving the home–school link, fostering healthy connections, increasing parental and educator competence in parenting and teaching methods, and producing positive outcomes for children. Because of their involvement in CBC, parents, teachers, and children have all shown signs of improvement in a number of randomized and controlled studies conducted in preschool. Parents and teachers who took part in CBC reported several gains in both their ability to solve problems and the quality of their relationships with one another. Like parents and teachers, participants experience some gains academically and behaviorally. Accordingly, social validity results are high for parents, teachers, and for participants. Parents and teachers conclude that the work being carried out in class caters to both their strengths and weaknesses, and teachers and CBC methods effectively support the participants to decrease their anxieties and well-being levels respectively.

This study concluded that the relationship that exists between a parent and their child’s teacher serves as a mediator in the influence that CBC has on teachers’ reports of the academic and behavioral difficulties that their pupils have. This finding is in line with what was discovered in earlier research, which indicates that CBC has a positive influence on the behavioral disorders that are exhibited by pupils [52,70,71]. More specifically, the progress in the relationship between the parent and the teacher has been partially accounted for by the impacts that CBC has had on some child outcomes. In addition to the impacts that are evident immediately, researchers have found that the effects of CBC are maintained throughout a one-year follow-up. These effects include parent reports of children’s academic and social skills as well as teacher reports of children’s difficulties in school [72]. Both of the photos in this study include the interventions and the baselines, which helps to understand the evidence indicating that the impacts of the interventions were almost immediately felt. Both of the photographs are included in this study. In addition, in order to ensure that the requirements of children with 2e are addressed within the context of the school environment, researchers have identified a large number of factors that should be taken into consideration in this respect. When teachers first acknowledge the student’s talents and then focus on fixing the student’s flaws, the student has a greater chance of achieving academic success, which in turn increases the potential for the teacher’s own academic success as well as the potential for that of the student [73]. This opinion is consistent with the procedures that were utilized in the research that was undertaken by the CBC, and therefore it is logical to take those methods into consideration.

## 5. Limitations

The conduct of research in preschool classrooms, where it may be difficult to establish high levels of control, comes with its own set of obstacles and limits that are unique to the setting [74,75]. According to Brown [75], a ‘trade-off between experimental control and richness and reality’ is necessary in order to make the transition toward educational settings more ecologically valid (p. 152). In this particular experiment, time constraints made it impossible to collect a significant number of prospective baseline observations and to carry out maintenance probes in an optimal manner, both of which required an immense degree of focus. Moreover, there is a lot of ‘noise’ in the classroom setting, such as interruptions to or abrupt changes in planned activities, alongside the unaccountable misconduct of other students. This ‘noise’ can have an impact on both the rigor of research and the actions of students while they are working on the tasks that have been assigned to them. This scenario, which at first appears to be a disadvantage, can, however, be interpreted as an advantage; it can be argued, for instance, that this hectic environment makes for a more authentic research setting [47].

The fidelity of this study was another area in which problems were encountered. When instructors were given the ability to pick how the strategy would be implemented, visual schedules were used in a variety of ways and were therefore perceived and applied subjectively. However, the participation of these teachers was essential to the overall objective of the study, which was to carry out an ecologically valid evaluation of these treatments. Kasari and Smith [47] highlight the importance of carrying out research in relevant settings in order to develop interventions that are capable of being implemented and maintained in real-world classrooms. Due to the fact that the research was conducted on a single sample, another shortcoming of the study is that it cannot be generalized to a larger population. Nevertheless, there is potential to address this in the future by conducting a larger quantity of similar experiments, from which more general deductions can be made.

## 6. Conclusions

It has been demonstrated that CBC has a positive influence on the issues of student behavior. To be more specific, the gains in the relationship between the parent and the teacher have been partially credited to the impacts that CBC has had on some child outcomes. This demonstrates that parents and teachers who have a good rapport with one another are more likely to be beneficial in the long run for supporting children who are taking part in CBC, in order for those children to achieve positive outcomes.

This study provided preliminary evidence that the use of visual schedules can increase participants’ ability to remain focused on the tasks at hand. As a result, the participants’ levels of annoyance and anxiety lessened once the process of being unable to attend to tasks was brought under control. When activities are carried out on the basis of visual schedule assistance and the concepts favored by the participants, it is considered that a sudden increase in on-task behavior is observed along with a synergistic effect between these two factors. Complaints about school, avoidance behaviors, violent responses, inattention, hyperactivity, and impulsivity all decreased, as well as withdrawal behaviors.

Subsequent research should seek to uncover whether the development of students with 2e is achieved through the application of visual support, or whether the creation of activities based on the strengths and interests of the students provides the required academic and behavioral development. In addition, it is essential to observe the long-term successes of students with 2e, thereby acknowledging that these children have particular social and emotional requirements, and to look for strategies that can assist these students in navigating the social dynamics of the school environment. Thus, keeping track of these children’s progress must be made an utmost priority in future studies.

## Figures and Tables

**Figure 1 behavsci-13-00674-f001:**
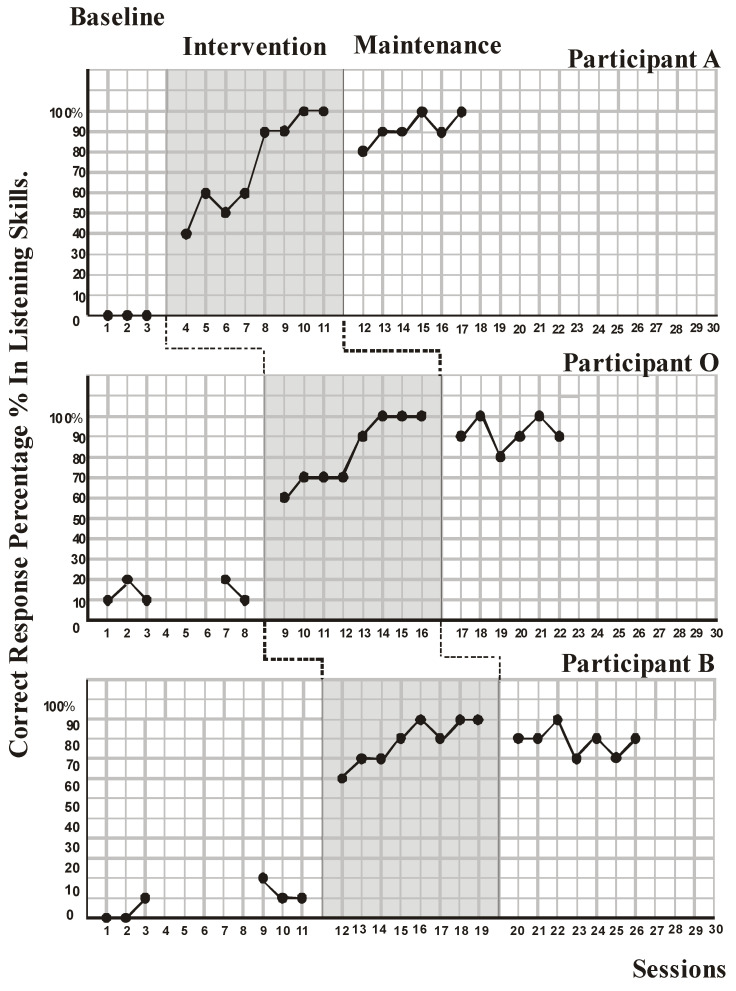
Correct response percentage in listening skills for Participant A, Participant O, and Participant B during the baseline, intervention, and maintenance sessions.

**Figure 2 behavsci-13-00674-f002:**
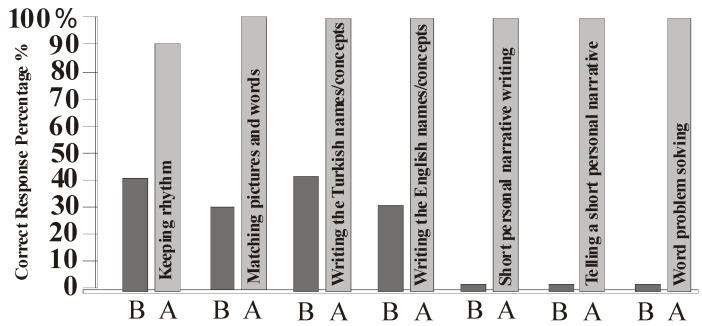
Total correct response percentage before and after intervention in related skills. B: before, A: after intervention.

**Figure 3 behavsci-13-00674-f003:**
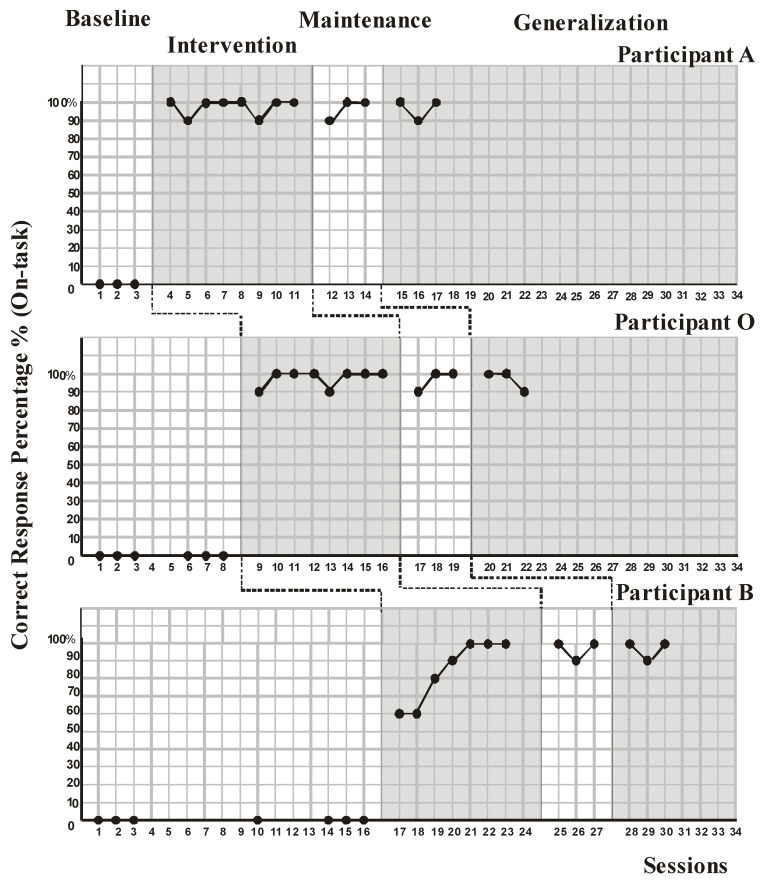
Correct response percentage in on-task behavior for Participant A, Participant O, and Participant B during the baseline, intervention, maintenance, and generalization sessions.

## Data Availability

Data will be available on the genuine request to the corresponding author.

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
