# Peer review of "Examining Conjoint Behavioral Consultation to Support 2e-Autism Spectrum Disorder and Gifted Students in Preschool with Academic and Behavior Concerns"

_behavsci, 2023, doi:10.3390/bs13080674_

Round 1

Reviewer 1 Report

The study used Conjoint Behavioral Consultation method to intervene three se-autistic children, which was meaningful. While the writing needs improving. In the Instruction part, the subtitles of 1.1 and 1.2 focused on teaching strategies for ASD, not highlighted related academic and behavior study, which made the academic and behavior indicators used in the study without powerful evidences. In Method and Materials part, section 2.2.2 & 2.2.3, the author listed many animal’s features which might not so necessary, one example is enough and others could be provide as Appendix. Furthermore, what’s the significant different between contents of section 2.2.2 and section 2.2.3?  As a whole, the article need major revision. The followings are some specific issues.

1.     The “listening skills” was an important variable used to reflect the intervention effects, but the article gave little information about how to assess it.

2.     In Finding part, only three graphs were displayed without any explained sentence and could not give clear results.

3.     There are three different “CBC”s in Page3, second paragraph. The first one might be the main concept from the article title and the whole text. But in Line 118. The author wrote “ CBC is an acronym for collaborative behavior assessment”. These cause confusion and it’s better not use acronyms of the other two concept.

4.     Page 11. Line517. The sentence “This was done by dividing the total number of times we saw the student by the total number of times we saw the student” was incomprehensible.

5.     Some of the writing format is not correct, such as the colons after titles on page 5 section 2.2.1. “Word problem samples:”, the use of semicolons in Page8. Line380-399 and Page10. Line481-486. The author should check the whole article carefully and make the necessary revises.

6.     The exact result of the fidelity test should be provided (section 2.8)

Author Response

In the attached file, referee 1's suggestions are explained point by point.

Reviewer 2 Report

I appreciate the topic and focus of the study, as well as a good outline of related theory - although, the outcomes have been expected and well supported by many previus reserach studies (regarding the structured teaching/learning at ASD children or the efficiency of teachers-parents'cooperation in education of 2e children, etc.).

In the first part, I suppose that since line 101 there should rather be another section focused specifically on CBC, including the new headline (as it looks like a part of 1.2 section at the moment, which is not very logical in the text structure...).

I miss the point "b)" - as following a) on line 119...

One can consider the methodology description to be a little bit confusing, as the research is referenced to preschool education (188) while there are some tasks assuming some typical school skills (reading, writing, maths) - see chapters 2.2.1 - 2.2.3. Generally, it is hard to expect and test these skills at preschool children (age five)...

The reference to particular research steps (phases) made by individual persons (authors...) engaged in this research are presented in a little bit confusing way as well (such details does not seem to be important if we consider the study a result of the team-work.

It seems that there are some particular mistakes in English grammar (e.g. see the question at line 175-175); sometimes there can a not well understandable sentence found (e.g. see lines 517-518) or there are some confusions in diacritics (see lines 550-551/563/692...).

Author Response

In the attached file, referee 2's suggestions are explained point by point.

Round 2

Reviewer 1 Report

The revised article was better with more clearly and logical structure. While, the language should be more concise. I still have a question, in page4, the last sententce of first paragrangh, "Additionally, the Canadian Broadcasting Corporation places an emphasis on creating healthy relationships between parents and their offspring", Here , what organization of "Canadian Broadcasting Corporation" ? Why the author emphasize it? Neither the corporation nor the whole sentence  could  be found  in referentce No.[33]

The writing could be more concise.

Author Response

The article was read by a native speaker editor. Since both reviewers provided minor suggestions on language, this editing is considered adequate.

The Canadian Broadcasting Corporation section has been removed from the article. It was considered unnecessary to provide additional explanation. The irrelevant reference has also been deleted. 1-2 sentences have been added to that paragraph (Please see Page 3, Line 143-146). This paragraph has been strengthened by adding a new reference (Please see Page 20, Line 876-879)
